# Effects of High-Intensity Ultrasound Pretreatment on Structure, Properties, and Enzymolysis of Walnut Protein Isolate

**DOI:** 10.3390/molecules27010208

**Published:** 2021-12-30

**Authors:** Fei Zhao, Xiaosong Zhai, Xuemei Liu, Meng Lian, Guoting Liang, Jingxiang Cui, Haizhou Dong, Wentao Wang

**Affiliations:** 1College of Agronomy and Environment, Shandong Facility Horticulture Bioengineering Research Center, Weifang University of Science and Technology, Weifang 262700, China; lianmeng@wfust.edu.cn (M.L.); lgt1984@wfust.edu.cn (G.L.); qiushui_920@163.com (J.C.); 2College of Food Science and Engineering, Engineering and Technology Center for Grain Processing of Shandong Province, Shandong Agricultural University, Taian 271018, China; xszhai@126.com (X.Z.); HaizhouDong888@sdau.edu.cn (H.D.); 3Jinan Fruit Research Institute, All-China Federation of Supply and Marketing Co-Operatives, Jinan 250014, China; liuxm0218@163.com

**Keywords:** high-intensity ultrasound, walnut protein isolates, particle size, zeta potential, fluorescence spectra, antioxidant activity

## Abstract

The purpose of this paper was to investigate the effect of high-intensity ultrasonication (HIU) pretreatment before enzymolysis on structural conformations of walnut protein isolate (WPI) and antioxidant activity of its hydrolysates. Aqueous WPI suspensions were subjected to ultrasonic processing at different power levels (600–2000 W) and times (5–30 min), and then changes in the particle size, zeta (ζ) potential, and structure of WPI were investigated, and antioxidant activity of its hydrolysates was determined. The particle size of the particles of aqueous WPI suspensions was decreased after ultrasound, indicating that sonication destroyed protein aggregates. The ζ-potential values of a protein solution significantly changed after sonication, demonstrating that the original dense structure of the protein was destroyed. Fourier transform infrared spectroscopy indicated a change in the secondary structure of WPI after sonication, with a decrease in β-turn and an increase in α-helix, β-sheet, and random coil content. Two absorption peaks of WPI were generated, and the fluorescence emission intensity of the proteins decreased after ultrasonic treatment, indicating that the changes in protein tertiary structure occurred. Moreover, the degree of hydrolysis and the antioxidant activity of the WPI hydrolysates increased after sonication. These results suggest that HIU pretreatment is a potential tool for improving the functional properties of walnut proteins.

## 1. Introduction

Walnuts (*Juglans regia* L.) are increasingly consumed for their nutritional attributes and health profile [1,2]. China and America are currently the two major walnut-producing countries [3]. Walnuts have high economic value due to the high level of lipids (67% on a dry basis) [4]. Lipids are rich in essential polyunsaturated fatty acids (particularly linoleic acid), which have nutritional advantages such as anti-oxidizing properties and the ability to lower blood cholesterol [5]. Walnuts contain about 18 to 24% proteins, which are very rich in essential amino acids [6]. Walnut proteins can be divided into four general categories: glutelin, globulin, albumin, and prolamin. The increasing market demand for walnut lipids has led to a large amount of a by-product: nutritional walnut proteins, which are used as animal feed or discarded [7]. It is necessary to increase the economic value of defatted walnut proteins, as the development of the walnut industry is discouraged by the underutilization of the by-product [5]. In particular, the hydrolysates of WPI have important biological activities including anti-atherogenic, anti-inflammatory, and anti-mutagenic properties [1,8]. However, the main group of proteins (about 70%) found in walnuts are glutelins, whose poor aqueous solubility limits their functional features as water-based food products.

To obtain ideal properties, many approaches are required to improve the physicochemical of walnut proteins, which can be used as functional foods or ingredients. Enzymatic modifications and chemical or physical methods were applied to protein processing [9,10]. High-intensity ultrasound (HIU) is an emerging non-thermal technique in food industries used to modify food properties and is often used for homogenization, filtration, and dehydration processes [11,12]. HIU uses high-energy mechanical waves (20–100 kHz) that induces cyclic generation and the collapse of cavities (sonication bubbles). The basic effects of HIU on liquid systems are mainly due to its ability to induce microstreaming currents and cavitation, followed by the formation of a local area of temperature and high pressure around the collapsed cavities, which can result in conformation changes to food proteins [13,14]. The physical alterations of the material properties are caused by the strong turbulence, cavitation, shear stresses, heating, and dynamic agitation generated through the ultrasonic process [15,16]. Recently, several researchers have proved the ability of sonication to improve the functional features of biomacromolecules, such as their emulsifying, interfacial, solubility, extraction, foaming, depolymerization, and gelling properties [10,17]. The foaming properties of soy protein isolate were improved by sonication, which was due to a change in the proteins’ molecular structure by ultrasonic waves [18]. The surface hydrophobicity and solubility of black-bean protein isolate were increased after sonication, which was ascribed to alterations in the proteins’ molecular structure [19].

Some studies found that ultrasonic treatment favors an increase in the degree of hydrolysis, which could alter the local structure of substrate proteins to make the active sites exposed [20,21]. Zhao et al. reported that the content of highly active antioxidant peptides increased in ultrasound-treated SPI hydrolysates because ultrasonication exposed the hydrophobic interaction sites of the proteins and increased the contact area between the substrates and enzyme [18]. However, little is known about the effect of sonication on the structure of walnut proteins or on the antioxidant activity of protein hydrolysates. To sufficiently understand the effect of HIU on the structure characteristics and properties of walnut proteins, particle size, zeta potential analyzer, Fourier transform infrared (FTIR) spectra, and fluorescence spectra were used. These results provided a new theoretical basis showing that HIU can be applied to improve the properties of proteins in the future.

## 2. Results and Discussion

### 2.1. Particle Size

The particle size of molecular aggregates formed in solutions can influence the functional properties of a protein [22,23]. The particle size of untreated control and sonicated WPI dispersions was measured (Table 1). Compared with the control (173.57 nm), there was a significant change (*p* 0.05) in the effective diameter (De) of the sonicated WPI samples (from 108.70 to 155.03 nm). The particle size of the protein samples was significantly reduced after sonication. Zhu et al. reported that sonication led to the narrowing of the particle size distribution in walnut protein dispersions and reduction in the particle diameter [10]. The decrease in effective diameter could be due to a disruption of some insoluble protein aggregates by the turbulent, cavitation, shear forces, and micro-streaming produced by the ultrasonic probe [24]. These results commonalities with earlier studies on whey proteins, sunflower protein isolates, fava bean protein, and soybean protein isolate [18,25,26,27]. Interestingly, the data indicated that over-processing of the WPI dispersions could lead to an increase in the particle size distribution of walnut protein dispersions. For instance, the effective diameter was higher when the protein samples were sonicated at 1200 W for 30 min than at 1200 W for 15 min, implying an increase in protein aggregation formation during ultrasonic processes. It may because the WPI denaturation became extensive after long-duration ultrasonic treatment, it actually promoted protein aggregation [10,18]. The changes in particle size caused by a longer sonication time are described as “over-processing” [28]. Zhou et al. also reported that over-processing of glycinin dispersions using sonication led to simultaneous dissociation and aggregation of the protein [29]. Aggregation of proteins may produce to ascribe noncovalent interactions, for instance, hydrophobic interactions, hydrogen bonds and electrostatic interactions [25]. The changes in the effective diameter of proteins could be attributed to hydrodynamic shear forces and cavitation of the ultrasonic probe, which disrupt associative electrostatic and hydrophobic interactions of the macromolecules [18]. The ultrasonic conditions could be optimized to ensure the availability of protein structure disruption, without promoting excessive denaturation of the proteins.

### 2.2. Effect of HIU Treatment on ζ-Potential of WPI

ζ-potential is an important physical and chemical index that shows the stability of a critical protein suspension solution. The ζ-potential depends on the surface charge characteristics of protein particles in a solution system [29]. Generally, a negative net charge is mainly due to glutamic and aspartic acid, and a positive net electronic charge is attributed to histidine and lysine acid [30]. If the negatively charged amino acids are exposed more than positively charged amino acids, the ζ-potential values of a protein solution are negative [31]. The production of various amino acid residues in proteins could form partial ionization, then the amino acid residues possess many surface negative charges, such as carboxyl groups [32]. Figure 1 presents the ζ-potential values of the WPI obtained after HIU treatment. The surface charges of the WPI significantly changed after sonication, and the highest ζ-potential (−10.43 mV) was obtained at 1400 W (20 min). The ζ-potentials of the samples treated at lower than 1800 W were negative, indicating that the more negatively charged groups were exposed on the protein WPI surface, while the ζ-potentials of the samples treated at 1800 W and 2000 W were positive. The change in absolute value of the ζ-potential of WPI suggested a change in the structural conformation and surface composition of the protein, which might be due to the exposure of charged moieties to the WPI surface proved by fluorescence spectroscopy, which is discussed in Section 2.4. In addition, the effective surface charge of WPI particles also mainly affects their aggregation of different sizes after ultrasonic treatment.

The ζ-potential value in a protein solution system was very important and reflected the system’s stability [33]. The aggregation or dispersion was easily shaped when the ζ-potential value was from −14 to −30 mV. The solution system represented mutual repulsion when the ζ-potential value was the greater than −30 mV and the system was stable enough [34]. The ζ-potential values of the WPI solution were the lower than −30 mV, and the solution system had poor physical stability. Ultrasonication (low and medium power) could increase the negative surface charged amino acids on proteins, strengthen the electrostatic repulsions between particles, destroy existing protein aggregates, inhibit further aggregation of proteins, and enhance the stability of the protein dispersions.

### 2.3. Effect of HIU on Secondary Structure of WPI

The FTIR spectra of proteins were recorded to reveal the changes in the secondary structure of WPI after sonication (Figure 2). The absorption bands of proteins with ultrasonic treatment were distinct from those of the untreated sample. The major spectral features were composed of three intense bands due to amide I (1656.92 cm^−1^), amide II (1537.40 cm^−1^), and amide III (1237.95 cm^−1^). The ultrasound-treated WPI had higher absorptive intensity than untreated WPI between 4000 and 400 cm^−1^. Sonication caused red shifts of amide I (about 1 cm^−1^) and amide II (about 1 cm^−1^) band peaks and a slight blue shift of the amide III band peaks. These results showed that the structural information of WPI was changed through sonication, in which the protein’s dense structure had been loosened during the sonication process, which can change the interactions between protein molecules and surface charge. A similar result was also reported by Martínez-Velasco et al. for fava beans [27].

The secondary structure information of proteins is generally based upon the absorption in amide I band analysis, where C=O (stretching vibration) has a main role, followed by bending of in-plane N-H, and by C-N (stretching modes) contributions [27]. Deconvolution of the amide I region was applied to separate and distinguish the secondary structures of WPI (Table 2). Gaussian peaks could be used for an analysis of their related structure based on the center. Table 2 shows the proportion of α-helix, β-sheet, β-turn, and random coil in the untreated WPI and sonicated protein samples. The results indicated that the secondary structure of WPI was significantly modified by sonication. In particular, the α-helix and random coil contents increased, while the β-sheet and β-turn content decreased with an increasing the intensity and processing time of sonication. The secondary structures of proteins are associated by different kinds of hydrogen bonds [35]. The amide I band of an IR spectrum is mainly ascribed to C=O stretching vibrations, which rely on various secondary structures and hydrogen bonds in molecules or between molecules [36]. These results suggest that sonication might have disrupted some kinds of hydrogen bonds, consequently resulting in some of the β-sheet and β-turn structures to be transformed into α-helix and random coil structures. Hu et al. found that SPI treated by higher power treatment (600 W) increased the α-helix and random coil [24]. Chandrapala et al. pointed out that sonication (60 min) showed an increase in the α-helix and a decrease in the β-turn structure of β-lactoglobulin [37].

Different changes in the secondary structure content of ultrasonic proteins were also monitored. For instance, the random coil content of WPI was reduced by HIU treated at 1200 W (10, 20, and 30 min), 1400 W (5 and 15 min), 1800 W (5, 25, and 30 min), and 2000 W (5, 15, and 30 min). Zhou et al. found that sonication influenced glycinin aggregates, while secondary structures were unchanged [29]. These diverse results may be due to the different sonication conditions and various native proteins. The shear forces generated by ultrasonic waves caused differences in the secondary structure and disrupted the interactions between protein molecules [38]. This could explain the differences in particle size in WPI induced by ultrasonic treatment due to exposure of hydrophobic and polar groups.

Sonication led to a red shift (about 19–68 cm^−1^) of amide A band peaks (3386.24 cm^−1^, N-H bending or O-H stretching vibration), indicating that more N-H groups of protein involved hydrogen bonding of polypeptide chains [39]. Sonication also caused red shifts of C-H_3_ asymmetric variable angle vibration (about 1–5 cm^−1^), C-C stretching (about 15–17 cm^−^^1^), and C=S stretching (about 1–3 cm^−1^), but caused a blue shift of amide VI (about 3–8 cm^−1^) band peaks and C-H_2_ asymmetric variable angle vibration (about 2 cm^−1^) [40]. Generally, shifting of the IR spectrum peaks indicated that the structures of the protein were unstable after HIU treatment. The auxochrome and chromophore groups of the protein were exposed after sonication, which include -OH, -NH_2_, -SH, C=O, N=N, and -COOR.

### 2.4. Fluorescence Spectra Analysis

The changes in the intrinsic fluorescence spectra of WPI solutions were provided as information on the structural changes in the proteins after ultrasonic treatment (Figure 3). When conformational changes in WPI occurred, the fluorescence spectra of a protein changed due to the local molecular environment of the tyrosine, phenylalanine, and tryptophan (Trp) groups [10]. The maximum wavelength (λ_max_) of fluorescence emission was around 338 nm for the untreated WPI. The resulting λ_max_ shown in Figure 3 shows a red shift with ultrasonic treatment. The red shift of all the samples showed that the protein structure had been loosened due to the aromatic amino acids exposed after ultrasonic treatment [41].

However, two absorption peaks of WPI were generated after ultrasonic treatment, and the fluorescence emission intensity of proteins decreased with increasing sonication time and intensity, which indicated that changes in protein aggregation and/or structure state occurred [42]. Sonication has been reported to decrease the fluorescence intensity of soy proteins and ovalbumin [43,44]. Presumably, HIU treatment altered the tertiary structure of the WPI, which caused a change in the local environment of the aromatic amino acid groups. Moreover, sonication can bring about a change to the proteins’ aggregation state and then influence the local environment of the groups. Therefore, there was a decrease in the fluorescence intensity of proteins according to the structural changes in in the proteins [45]. HIU uses energy mechanical waves that produces a collapse of cavities and cyclic generation followed by the formation of a localized region of temperature and high pressure surrounding the collapsed cavities, which can induce conformational changes in the proteins [13]. The results were consistent with the Fourier transform infrared spectroscopy and particle size measurements, which suggested that HIU caused changes in the protein structure.

### 2.5. Degree of Hydrolysis of WPI Hydrolysates

As can be seen in Table 3, the DH of the WPI hydrolysates was changed significantly (*p* 0.05) after ultrasonic treatment compared to that of the untreated protein hydrolysates, and the maximum DH of the samples treated at 1200 W for 10 min was about 19.83%. Compared with the control, the DH of the WPI-1200 W-10 min, WPI-1000 W-5 min, and WPI-600 W-15 min hydrolysates increased by 67.5%, 57.5%, and 52.5%, respectively. The different DH of the WPI hydrolysates further confirmed the influence of ultrasonic treatment on the particle size, surface charge properties, and structure of the protein. These results suggested that the microstreaming currents and cavitation of ultrasonic waves altered the surface structure and charge of WPI, and the enzymes’ sensitive sites of protein fractions were exposed [46]. The contact area between enzymes and substrates was increased due to the exposure of the enzymes’ sensitive sites in protein fractions, and the extent of subsequent proteolysis was improved [20]. The WPI-1200 W-10 min hydrolysates obtained the highest DH, while the WPI-1200 W-15 min hydrolysates had the highest hydroxide radical-scavenging activity (55.97%) and the WPI-800 W-20 min hydrolysates had the highest DPPH radical-scavenging activity (86.95%). The results suggested that the antioxidant activity of the protein hydrolysates appeared to be correlated to a certain degree with the DH of WPI, but the hydrolysates with higher DH did not have higher antioxidant activity. This was probably because the antioxidant peptides of the protein hydrolysates with a higher DH were broken up into amino acids or smaller peptides, then their antioxidant activity decreased [18,47].

### 2.6. Antioxidant Activity of WPI Hydrolysates

The antioxidant potential of WPI hydrolysates was determined by the scavenging activity of hydroxide radicals and DPPH radicals. A higher absorbance indicates a higher hydroxide and DPPH radical-scavenging activity (*p* 0.05). Table 4 and Table 5 show the sonication power level of WPI, as well as the ultrasonic processing time, both of which influenced the antioxidant effects of the WPI hydrolysates. The WPI hydrolysates treated at 1200 W for 15 min revealed the highest hydroxide radical-scavenging activity (55.97%) in all samples (Table 4). Compared with the control, the hydroxide radical-scavenging activity of the WPI-600 W-5 min and WPI-800 W-15 min hydrolysates increased by 20.48% and 19.32%, respectively. The differences in DPPH and hydroxide radical-scavenging activity of the WPI hydrolysates were significant. As can be seen from the Table 5, the WPI hydrolysates exhibited various scavenging activities against DPPH radicals. The higher DPPH radical-scavenging activity (86.95%) of the WPI hydrolysates was obtained when the protein endured HIU treatment (800 W, 20 min). The DPPH radical-scavenging activity of the protein hydrolysates treated at 800 W (5, 10, 20, and 30 min), 1200 W (5, 15, 20, 25, and 30 min), 1400 W (5, 10, and 15 min), and 1800 W (20 min) was higher than that of the control (80.6%), while the antioxidant activity of the other ultrasonic-treated protein hydrolysate was lower than that of the control. These results may be due to the structural changes in WPI fractions after sonication, which caused the antioxidant activity of the WPI hydrolysates to be different. It is worth noting that the antioxidant activity of the WPI hydrolysates (5 mg/mL) was higher than that of the SPI hydrolysates (5 mg/mL) in our previous studies [18], which suggest that WPI peptide had higher antioxidant activity. HIU pretreatment could alter the local environment of the protein groups, and the hydrophobic interaction sites of WPI were exposed; subsequently, increasing the contact area between the enzyme and substrates improved the antioxidant activity of the WPI hydrolysates. With the increase in ultrasonic time, the protein hydrophobic interaction sites, which were obscured inside the molecules, were exposed to the surface [39,48]. Enzymatic hydrolysis of proteins combined with ultrasonic pretreatment could be beneficial for obtaining effective active peptides. The results showed that protein peptides with high antioxidant activity were obtained by low-power ultrasonic treatment, and WPI peptides with higher antioxidant activity depended on suitable ultrasonic treatment conditions.

## 3. Materials and Methods

### 3.1. Materials

Walnut protein isolate (WPI; protein content 80.31%) was provided by Shandong Wonderful Industrial Group Co., Ltd. (Dongying, China). The hydrolysis of WPI was performed using food-grade bromelain (E.C. 3.4.22.32, Guangxi Pangbo Biological Engineering Co., Ltd., Nanning, China) with an enzyme activity of 500,000 U/g.

### 3.2. HIU Treatment of WPI

To prepare WPI (5% *w/v*) solutions, WPI powder dissolved in deionized water was stirred for 2 h (room temperature) and kept refrigerated. An ultrasound processor (TJS-3000 Intelligent Ultrasonic Generator V6.0, 20 kHz, 25 mm diameter titanium probe, Hangzhou Success Ultrasonic Equipment Co., Ltd., Hangzhou, China) was used to process 50 mL of WPI solutions in 100 mL glass vessels that were immersed in an ice-water bath, and ice was added every 5 min. An ultrasound processor (TJS-3000 Intelligent Ultrasonic Generator V6.0, 20 kHz, 25 mm diameter titanium probe, Hangzhou Success Ultrasonic Equipment Co., Ltd., Hangzhou, China) was used for the preparation of WPI solutions (50 mL) in glass vessels (100 mL) that were immersed in an ice-water bath. Protein solutions were treated at 600 W, 800 W, 1000 W, 1200 W, 1400 W, 1600 W, 1800 W, and 2000 W for 0, 5, 10, 15, 20, 25, and 30 min. Samples were stored for further use after lyophilization.

### 3.3. Particle Size and Zeta Potential Measurements

The particle size distribution of the WPI samples was measured following HIU treatment. Before the test, protein samples (1.5 mg/mL) were filtered with Millipore filters (0.45 μm) to remove dust. The effective diameter (De) of all samples was measured by a zeta potential analyzer (Brookhaven Instruments, Holtsville, NY, USA) with a refraction index of 1.334. The ζ potentials of the samples were determined using a zeta potential analyzer. The sample solutions were transferred into the cuvette for 30 min before measurements. The ζ-potential value was the average of three measurements.

### 3.4. Fourier Transform Infrared Spectroscopy

The IR absorption spectrum of all samples was recorded with an FTIR spectrometer (Nicolet is5, Thermo Scientific, Waltham, MA, USA). All samples were prepared into potassium bromide pellets, and then a spectrum of samples was recorded in the range of 4000–500 cm^−1^. The resolution and scan number were 4 cm^−1^ and 32/sample, respectively.

### 3.5. Fluorescence Measurements

The fluorescence spectra of HIU-treated and untreated protein samples were obtained by a Lumina fluorescence spectrometer (Thermo Fisher Scientific, Waltham, MA, USA). The protein samples were dissolved in phosphate buffer (10 mM, pH 7.0) to obtain a concentration of 0.2 mg/mL. The protein solution was excited at a wavelength of 290 nm and the fluorescence emission was recorded from between 300 and 500 nm at a bandwidth of 5 nm for both excitation and emission.

### 3.6. Preparation of WPI Hydrolysates

Untreated and HIU-treated WPI dispersions (5% *w/v*) were dissolved in distilled water. The WPI solutions were incubated in a water bath at 55 °C and adjusted to pH 7.0 with 1 M NaOH during the reaction. The WPI solutions were subjected to enzymatic hydrolysis by bromelain (1:20 enzyme/substrate ratio). The enzyme was inactivated by boiling water for 10 min after hydrolysis, and then the enzymolysis liquid was centrifuged at 10,000× *g* for 10 min. The supernatant was collected, freeze-dried, and stored for further analysis.

### 3.7. Degree of Hydrolysis

The degree of hydrolysis (DH) of walnut protein was obtained from the pH-stat method [49]. The DH of the sample was calculated according to the following formula:(1)$$\mathrm{DH}=\frac{B\times N}{\alpha \times M_{p}\times h}\times 100\%$$
where B is the NaOH consumption (mL); N is the concentration of NaOH (1 M); α is the degree of α-amino groups dissociation; M_p_ is the protein mass (g); and for WPI, h = 7.75 mmol/g protein.

### 3.8. Determination of Antioxidant Activity of WPI Hydrolysates

#### 3.8.1. Hydroxyl Radical-Scavenging Capacity Assay

The reaction mixture containing the sample solution (2 mL, 5 mg/mL), FeSO_4_ (2 mL, 6 mM), and H_2_O_2_ (2 mL, 6 mM) solutions was allowed to sit at 30 °C for 10 min, then the mixture with the addition of salicylic acid (2 mL, 6 mM) was incubated at 37 °C for 30 min [50]. The absorbance at 510 nm was measured with a UV–Visible spectrophotometer (UV-8000 A, Analytical Instrumental, Shanghai, China). The hydroxyl radical-scavenging capacity of the sample was calculated using the following equation:(2)$$\mathrm{Hydroxyl}\text{-}\mathrm{radical}\;\mathrm{scavenging}\;\mathrm{capacity}\;(\%)=(1-\frac{A_{i}-A_{j}}{A_{0}})\times 100\%$$
where A_i_ was the absorbance of FeSO_4_, H_2_O_2_, and salicylic acid with the WPI hydrolysates; A_j_ was the absorbance of FeSO_4_, H_2_O_2_, and distilled water with the WPI hydrolysates; and A_0_ was the absorbance of FeSO_4_, H_2_O_2_, and salicylic acid with distilled water.

#### 3.8.2. DPPH Scavenging Activity

The DPPH radical-scavenging activity of the hydrolysates was determined by the method of Yamaguchi et al. [51]. A total of 2 mL of the sample hydrolysate solution (5 mg/mL) was mixed with 2 mL of 20 μM ethanolic DPPH solution. The mixture was shaken and incubated at 30 °C for 30 min in the dark. The absorbance of the sample solution was determined at 517 nm with a UV–Visible spectrophotometer (UV-8000 A, Analytical Instrumental, Shanghai, China). The scavenging activity of the WPI hydrolysates was calculated using the equation:(3)$$\mathrm{DPPH}\;\mathrm{scavenging}\;\mathrm{activity}\;(\%)=(1-\frac{A_{1}-A_{2}}{A_{3}})\times 100\%$$
where A_1_ is the absorbance of the WPI hydrolysate with DPPH, A_2_ is the absorbance of the WPI hydrolysate with ethanol, and A_3_ is the absorbance of distilled water with DPPH.

### 3.9. Data Analysis

The data were presented as the means ± standard deviations (SD) of three replicate determinations. Statistical analysis was performed using SPSS (20.0) software. Significant differences (*p* 0.05) between means were identified using Duncan’s multiple range test. The analysis of the Fourier transform infrared spectroscopy and fluorescence spectra was conducted using OMNIC software (Thermo Fisher Scientific, Waltham, MA, USA) and Peak Fit software version 4.0 (Systat Software Inc., San Jose, CA, USA).

## 4. Conclusions

The HIU pretreatment changed the structure of WPI and antioxidant activity of its hydrolysates. Compared to the control WPI, ultrasonic processing decreased particle size and changed the surface charge characteristics and structure, which subsequently exposed the enzymes’ sensitive sites of protein fractions. The DH, hydroxide, and DPPH radical-scavenging activity of the protein hydrolysates increased in the ultrasound-treated WPI. The effects were ascribed to the function of the ultrasonic waves disrupting the structure of globular protein molecules, resulting in the partial unfolding of the protein; hence, the WPI was more prone to being hydrolyzed. The results could be significant for improving the utilization of WPI as natural plant functional proteins.

## Figures and Tables

**Figure 1 molecules-27-00208-f001:**
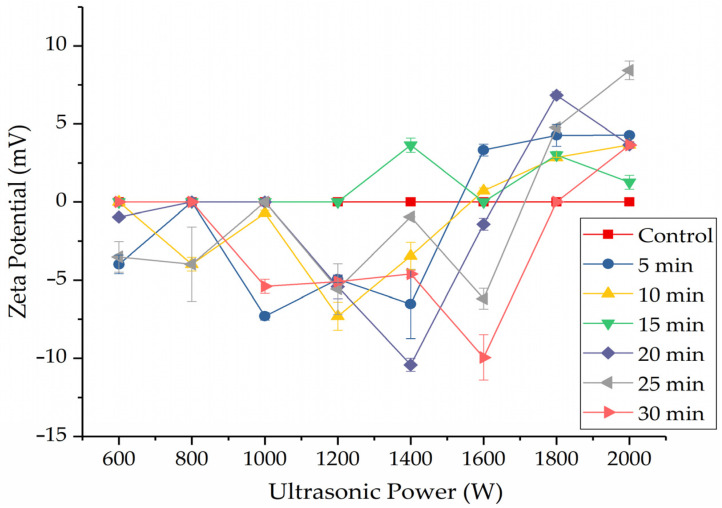
ζ-potential value of HIU–treated WPI.

**Figure 2 molecules-27-00208-f002:**
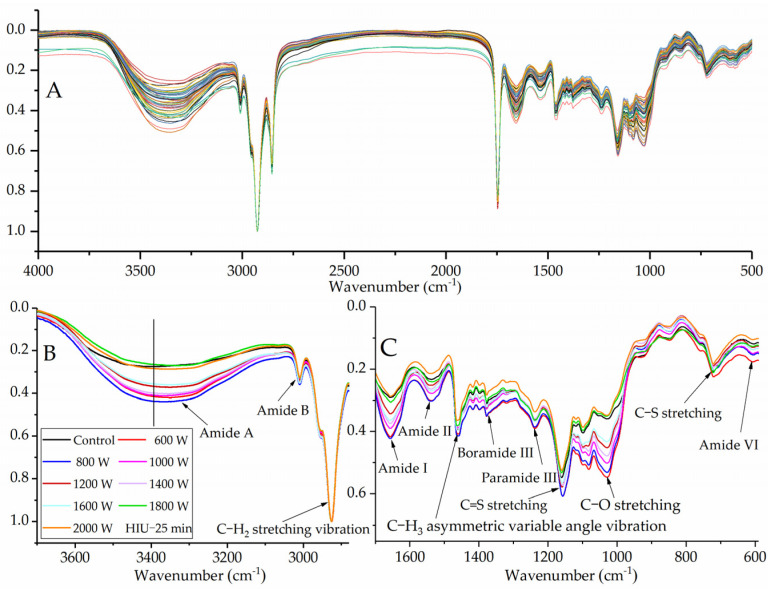
FTIR spectra of WPI after HIU treatment ((**A**): 4000–500 cm^−1^) and HIU treatment for 25 min ((**B**): 3400–2800 cm^−1^, (**C**): 1800–600 cm^−1^).

**Figure 3 molecules-27-00208-f003:**
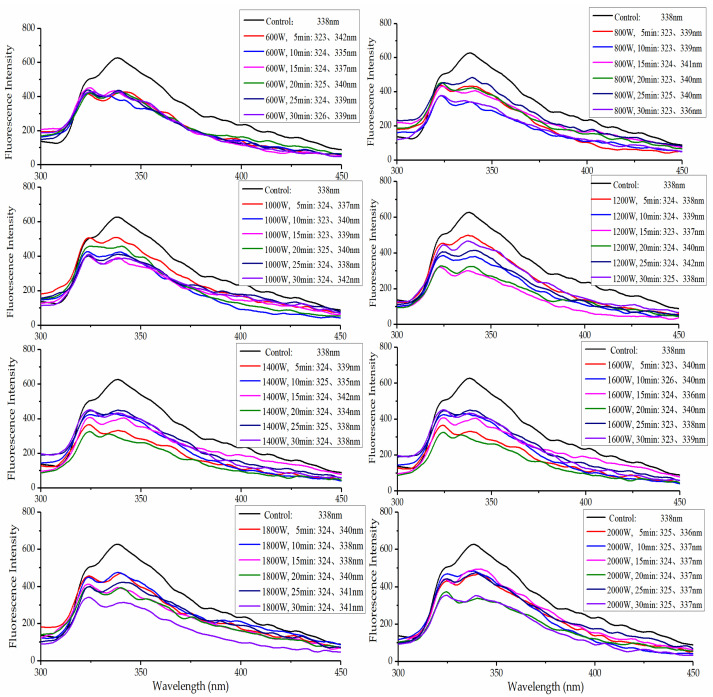
Fluorescence spectra of WPI after HIU treatment.

**Table 1 molecules-27-00208-t001:** Effects of HIU treatment on the effective diameter of WPI.

Ultrasonic Power (W)	Ultrasonic Processing Time (min)						
	0	5	10	15	20	25	30
	Effective Diameter (nm)						
600	173.57 ± 0.15 ^Aa^	143.03 ± 0.70 ^Bf^	134.33 ± 1.12 ^Cd^	128.87 ± 0.31 ^Dd^	134.60 ± 1.41 ^Ccd^	133.83 ± 2.20 ^Cd^	124.90 ± 1.32 ^Eg^
800	173.57 ± 0.15 ^Aa^	149.57 ± 0.90 ^Bd^	129.70 ± 0.75 ^Ee^	144.37 ± 2.06 ^Cb^	136.97 ± 2.48 ^Dc^	130.90 ± 0.62 ^Ee^	144.90 ± 0.44 ^Cc^
1000	173.57 ± 0.15 ^Aa^	140.90 ± 1.66 ^Cg^	129.40 ± 0.26 ^Ee^	128.07 ± 1.60 ^Ede^	135.07 ± 2.14 ^Dcd^	143.80 ± 1.21 ^Bc^	141.00 ± 0.60 ^Cd^
1200	173.57 ± 0.15 ^Aa^	147.27 ± 1.04 ^Ce^	143.90 ± 0.50 ^Db^	127.17 ± 0.59 ^Fe^	140.10 ± 1.61 ^Eb^	115.63 ± 0.55 ^Gg^	155.03 ± 0.29 ^Bb^
1400	173.57 ± 0.15 ^Aa^	138.67 ± 0.35 ^Dh^	108.70 ± 0.20 ^Gf^	117.03 ± 0.15 ^Ef^	114.63 ± 0.12 ^Fe^	148.57 ± 0.35 ^Bb^	144.40 ± 0.53 ^Cc^
1600	173.57 ± 0.15 ^Aa^	155.20 ± 0.26 ^Bb^	130.13 ± 0.71 ^Ee^	112.03 ± 0.51 ^Gg^	116.03 ± 0.29 ^Fe^	143.43 ± 0.38 ^Cc^	136.80 ± 0.95 ^De^
1800	173.57 ± 0.15 ^Aa^	152.60 ± 0.62 ^Bc^	139.93 ± 0.42 ^Cc^	133.17 ± 0.06 ^Ec^	133.87 ± 0.15 ^Dd^	126.00 ± 0.36 ^Gf^	127.00 ± 0.10 ^Ff^
2000	173.57 ± 0.15 ^Aa^	150.67 ± 0.32 ^Bd^	144.60 ± 0.36 ^Cb^	127.27 ± 0.15 ^Edc^	136.83 ± 0.15 ^Dc^	114.33 ± 0.12 ^Gg^	124.60 ± 0.72 ^Fg^

Data are the averages of three replications ± standard deviation. A–G: Different uppercase letters mean significant difference in comparisons between ultrasonic processing time (0, 5, 10, 15, 20, 25, and 30 min) for each power at *p* 0.05. a–g: Different lowercase letters mean significant differences between ultrasonic power (600, 800, 1000, 1200, 1400, 1600, 1800, and 2000 W) for each processing time at *p* 0.05.

**Table 2 molecules-27-00208-t002:** Effects of HIU treatment on the secondary structure composition of WPI.

Entry	α-Helix (%)	Random Coil (%)	β-Sheet (*%*)	β-Turn (*%*)
Control	7.55	15.80	31.01	45.64
600 W-5 min	6.54	15.89	32.00	45.57
600 W-10 min	8.92	15.06	31.07	44.95
600 W-15 min	8.92	15.05	31.00	45.03
600 W-20 min	7.14	15.63	31.90	45.33
600 W-25 min	7.05	15.63	31.66	45.66
600 W-30 min	8.69	15.21	30.36	45.74
800 W-5 min	6.84	16.28	31.57	45.32
800 W-10 min	8.46	15.53	30.61	45.4
800 W-15 min	7.23	15.66	31.65	45.45
800 W-20 min	7.23	15.58	31.75	45.44
800 W-25 min	6.93	15.61	32.21	45.24
800 W-30 min	8.81	15.04	31.26	44.89
1000 W-5 min	8.09	15.05	31.63	45.23
1000 W-10 min	6.58	15.97	32.17	45.28
1000 W-15 min	7.50	15.84	31.03	45.62
1000 W-20 min	7.45	15.73	31.50	45.32
1000 W-25 min	7.86	14.99	33.16	43.98
1000 W-30 min	7.96	15.12	32.66	44.29
1200 W-5 min	8.11	14.42	33.54	43.93
1200 W-10 min	7.56	15.89	31.17	45.38
1200 W-15 min	8.59	14.88	32.17	44.37
1200 W-20 min	8.01	15.13	32.47	44.39
1200 W-25 min	7.85	14.11	33.18	44.85
1200 W-30 min	8.65	14.75	32.12	44.47
1400 W-5 min	7.41	15.79	31.52	45.28
1400 W-10 min	8.32	14.47	32.68	44.53
1400 W-15 min	7.90	15.04	32.62	44.44
1400 W-20 min	6.84	16.08	31.24	45.84
1400 W-25 min	8.30	15.38	31.36	44.96
1400 W-30 min	7.34	15.75	32.01	44.9
1600 W-5 min	8.44	15.55	30.93	45.08
1600 W-10 min	6.81	15.92	31.57	45.69
1600 W-15 min	6.56	15.84	31.87	45.73
1600 W-20 min	7.43	15.59	31.6	45.37
1600 W-25 min	8.75	14.93	31.36	44.96
1600 W-30 min	8.55	14.98	30.80	45.67
1800 W-5 min	7.46	15.74	31.17	45.64
1800 W-10 min	7.36	15.58	31.51	45.55
1800 W-15 min	7.48	15.65	30.97	45.90
1800 W-20 min	7.87	14.96	32.70	44.48
1800 W-25 min	7.90	15.00	32.66	44.44
1800 W-30 min	8.37	14.64	33.03	41.76
2000 W-5 min	7.97	15.11	32.56	44.36
2000 W-10 min	7.82	14.95	32.95	44.28
2000 W-15 min	8.09	15.15	32.18	44.58
2000 W-20 min	6.26	15.60	33.01	45.12
2000 W-25 min	8.00	15.17	32.65	44.18
2000 W-30 min	7.55	15.80	31.01	45.64

**Table 3 molecules-27-00208-t003:** Effect of HIU pretreatment on degree of hydrolysis of the WPI hydrolysates.

Ultrasonic Power (W)	Ultrasonic Processing Time (min)						
	0	5	10	15	20	25	30
	Degree of Hydrolysis (%)						
600	11.84 ± 0.51 ^Da^	14.50 ± 0.51 ^BCd^	13.91 ± 0.51 ^Ce^	18.05 ± 0.51 ^Aa^	13.61 ± 0.51 ^Cc^	15.09 ± 0.00 ^Bc^	14.50 ± 0.51 ^BCc^
800	11.84 ± 0.51 ^Ca^	10.95 ± 0.51 ^Df^	11.07 ± 0.45 ^Dg^	15.68 ± 0.51 ^Bbc^	12.55 ± 0.21 ^Ce^	16.57 ± 0.51 ^Ab^	10.06 ± 0.51 ^Ee^
1000	11.84 ± 0.51 ^Da^	18.64 ± 0.00 ^Aa^	18.05 ± 0.51 ^Ab^	16.28 ± 0.51 ^Bb^	14.50 ± 0.51 ^Cc^	16.69 ± 0.31 ^Bb^	18.05 ± 0.51 ^Aa^
1200	11.84 ± 0.51 ^Ea^	18.20 ± 0.44 ^Ba^	19.83 ± 0.51 ^Aa^	15.39 ± 0.51 ^Dc^	18.05 ± 0.51 ^Ba^	16.39 ± 0.45 ^Cb^	16.39 ± 0.45 ^Cb^
1400	11.84 ± 0.51 ^Ea^	14.50 ± 0.51 ^Cd^	17.16 ± 0.51 ^Ac^	12.72 ± 0.51 ^Dc^	16.28 ± 0.51 ^Bb^	14.32 ± 0.21 ^Cd^	14.50 ± 0.51 ^Cc^
1600	11.84 ± 0.51 ^Ea^	16.28 ± 0.51 ^Ab^	14.91 ± 0.31 ^Bd^	14.32 ± 0.21 ^Cd^	14.20 ± 0.36 ^Cc^	13.49 ± 0.31 ^De^	16.10 ± 0.21 ^Ab^
1800	11.84 ± 0.51 ^Ea^	15.27 ± 0.31 ^Bc^	14.32 ± 0.21 ^Cd^	15.51 ± 0.45 ^Bbc^	13.91 ± 0.51 ^Cc^	17.87 ± 0.21 ^Aa^	12.72 ± 0.51 ^Dd^
2000	11.84 ± 0.51 ^Ea^	12.55 ± 0.21 ^De^	12.43 ± 0.00 ^Df^	13.20 ± 0.21 ^Ce^	14.38 ± 0.31 ^Bc^	15.21 ± 0.21 ^Ac^	14.15 ± 0.10 ^Bc^

Data are the averages of three replications ± standard deviation. A–G: Different uppercase letters mean significant difference in comparisons between ultrasonic processing time (0, 5, 10, 15, 20, 25, and 30 min) for each power at *p* 0.05. a–g: Different lowercase letters mean significant differences between ultrasonic power (600, 800, 1000, 1200, 1400, 1600, 1800, and 2000 W) for each processing time at *p* 0.05.

**Table 4 molecules-27-00208-t004:** Effect of HIU treatment on hydroxyl radical-scavenging of the WPI hydrolysates.

Ultrasonic Power (W)	Ultrasonic Processing Time (min)						
	0	5	10	15	20	25	30
	Hydroxyl Radical-Scavenging Activity (%)						
600	43.92 ± 0.25 ^Ea^	55.76 ± 0.38 ^Aa^	54.65 ± 0.32 ^Ba^	49.95 ± 0.86 ^Dc^	48.46 ± 0.68 ^Ec^	51.01 ± 0.82 ^Cb^	53.82 ± 0.57 ^Ba^
800	43.92 ± 0.25 ^Ea^	52.37 ± 0.22 ^Bb^	51.32 ± 0.73 ^Cc^	55.22 ± 0.79 ^Aa^	50.28 ± 0.28 ^Dc^	46.85 ± 0.44 ^Ed^	46.89 ± 0.61 ^Ee^
1000	43.92 ± 0.25 ^Ea^	48.54 ± 0.39 ^Ce^	42.77 ± 0.36 ^Ff^	51.34 ± 0.33 ^Bb^	53.08 ± 1.00 ^Aa^	46.62 ± 0.74 ^Dd^	40.41 ± 0.33 ^Gf^
1200	43.92 ± 0.25 ^Ea^	49.19 ± 0.35 ^Cd^	48.69 ± 0.20 ^Cd^	55.97 ± 0.64 ^Aa^	48.65 ± 0.51 ^Cd^	54.55 ± 0.68 ^Be^	48.28 ± 0.57 ^Cd^
1400	43.92 ± 0.25 ^Ea^	44.03 ± 0.56 ^Dg^	41.11 ± 0.54 ^Eg^	48.25 ± 0.36 ^Bd^	48.09 ± 0.24 ^Bd^	46.41 ± 0.60 ^Cc^	50.43 ± 0.51 ^Ac^
1600	43.92 ± 0.25 ^Ea^	49.76 ± 0.48 ^Dd^	54.14 ± 0.47 ^Aa^	48.81 ± 0.35 ^Dc^	51.27 ± 0.58 ^Cb^	50.47 ± 0.32 ^Cb^	53.02 ± 0.86 ^Ba^
1800	43.92 ± 0.25 ^Ea^	47.80 ± 0.66 ^Cf^	43.89 ± 0.79 ^Ee^	45.69 ± 0.91 ^Dc^	50.15 ± 0.48 ^Bc^	48.85 ± 0.18 ^Cc^	52.25 ± 0.63 ^Ab^
2000	43.92 ± 0.25 ^Ea^	50.55 ± 0.46 ^Bc^	52.59 ± 0.68 ^Ab^	46.11 ± 0.19 ^De^	46.28 ± 0.79 ^De^	46.59 ± 0.43 ^Dd^	48.95 ± 0.46 ^Cd^

Data are the averages of three replications ± standard deviation. A–G: Different uppercase letters mean significant difference in comparisons between ultrasonic processing time (0, 5, 10, 15, 20, 25, and 30 min) for each power at *p* 0.05. a–g: Different lowercase letters mean significant differences between ultrasonic power (600, 800, 1000, 1200, 1400, 1600, 1800, and 2000 W) for each processing time at *p* 0.05.

**Table 5 molecules-27-00208-t005:** Effect of HIU treatment on DPPH radical-scavenging of the WPI hydrolysates.

Ultrasonic Power (W)	Ultrasonic Processing Time (min)						
	0	5	10	15	20	25	30
	DPPH Radical-Scavenging Activity (%)						
600	80.63 ± 0.40 ^Aa^	75.92 ± 0.61 ^Ce^	75.98 ± 0.65 ^Ce^	75.83 ± 0.51 ^Ce^	77.89 ± 0.20 ^Be^	77.49 ± 0.64 ^Bd^	77.92 ± 0.82 ^Bd^
800	80.63 ± 0.40 ^Ba^	86.10 ± 0.16 ^Aa^	86.47 ± 0.77 ^Aa^	80.85 ± 0.71 ^Bc^	86.95 ± 0.25 ^Aa^	80.30 ± 0.49 ^Bb^	86.32 ± 0.57 ^Aa^
1000	80.63 ± 0.40 ^Aa^	78.69 ± 0.56 ^Bd^	78.12 ± 0.26 ^Bc^	80.80 ± 0.84 ^Ac^	80.58 ± 0.37 ^Ac^	80.04 ± 0.22 ^Ab^	77.76 ± 0.66 ^Bd^
1200	80.63 ± 0.40 ^Da^	83.06 ± 0.57 ^Bb^	79.30 ± 0.36 ^Ec^	82.39 ± 0.37 ^Bb^	81.54 ± 0.44 ^Cb^	81.38 ± 0.30 ^Ca^	84.56 ± 0.58 ^Ab^
1400	80.63 ± 0.40 ^Ba^	80.93 ± 0.36 ^Bc^	81.01 ± 0.55 ^Bb^	84.24 ± 0.67 ^Aa^	78.52 ± 0.11 ^Cd^	79.21 ± 0.08 ^Cc^	78.57 ± 0.70 ^Cc^
1600	80.63 ± 0.40 ^Aa^	76.71 ± 0.52 ^De^	79.12 ± 0.65 ^Bc^	75.56 ± 0.75 ^De^	77.55 ± 0.90 ^Ce^	75.96 ± 0.62 ^De^	77.96 ± 0.17 ^Cd^
1800	80.63 ± 0.40 ^Ba^	75.74 ± 0.36 ^Ee^	76.11 ± 0.50 ^Ee^	75.67 ± 0.88 ^Ee^	82.20 ± 0.17 ^Ab^	77.16 ± 0.35 ^Dd^	79.07 ± 0.57 ^Cc^
2000	80.63 ± 0.40 ^Aa^	78.11 ± 0.33 ^Bd^	76.19 ± 0.61 ^Ce^	78.36 ± 0.53 ^Bd^	77.00 ± 0.23 ^Cf^	80.13 ± 0.78 ^Ab^	77.00 ± 0.21 ^Ce^

Data are the averages of three replications ± standard deviation. A–G: Different uppercase letters mean significant difference in comparisons between ultrasonic processing time (0, 5, 10, 15, 20, 25, and 30 min) for each power at *p* 0.05. a–g: Different lowercase letters mean significant differences between ultrasonic power (600, 800, 1000, 1200, 1400, 1600, 1800, and 2000 W) for each processing time at *p* 0.05.

## Data Availability

Not applicable.

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
