# Peer review of "Effects of High-Intensity Ultrasound Pretreatment on Structure, Properties, and Enzymolysis of Walnut Protein Isolate"

_molecules, 2021, doi:10.3390/molecules27010208_

Round 1

Reviewer 1 Report

Comments in the attached file. 

Author Response

Responses to the review comment on “High-intensity ultrasound pretreatment before enzymolysis of walnut protein isolate: effect of ultrasound on structural conformations and functional properties of the protein” (ID: molecules-1481860). Dear editor, Thank you very much for giving us an opportunity to revise our manuscript again, we appreciate editor and reviewers very much for their positive and constructive comments on our manuscript entitled “High-intensity ultrasound pretreatment before enzymolysis of walnut protein isolate: effect of ultrasound on structural conformations and functional properties of the protein” (ID: molecules-1481860). We have studied reviewer’s comments carefully and made revision which marked in red in the paper according to the comments. Now we submit our revised manuscript for your kind consideration. I am the corresponding author for this paper. If you have any questions, please let me know. I am looking forward to hearing from you soon. Thank you very much for your kind help. Yours sincerely, Fei Zhao & Wentao WangTel: +86 53 8242850 E-mail address: feizhaozhaofei@126.com (F.Z.), wwtlxm@126.com (W.T.W.) 1.The title is unclear because structure properties are only performed on the non-hydrolyzed protein, whereas hydrolysis degree and antioxidant properties were executed on the enzyme-treated protein. In addition, the title implies that all properties were tested before enzymolysis. Therefore, I suggest changing the title. Response: The title has been revised. 2.Abstract. Clarify the meaning of DH. Response: The meaning of DH has been clarified. 3.Introduction. Scientific names as Juglans regia must be in italics. Response: “Juglans regia” has been in italics. 4.Materials and Methods. Lines 92-95 information is twice and for SPI instead of WPI. Response: SPI has been replaced by WPI. 5.Materials and Methods. Protein treatments indicate the percentage of power; it is better to put this information in watts as in the rest of the document. Response: The section has been revised in the revised manuscript. The power of protein treatments has been uniformly converted to watts. 6.Materials and Methods. The particle size and zeta potential subsections could be in one and not two sections because the same equipment is used to obtain both properties. Response: The particle size and zeta potential subsections have been integrated into “Section 2.3” in the revised manuscript. 7.In the case of preparation of WPI hydrolysates, how the high temperature for enzyme inactivation could affect the functionality evaluated in the product?. Response: The common method of enzyme inactivation is boiling water inactivation by reviewing the literature, which is performed under the same controlled conditions. The effect of high temperature for enzyme inactivation on the functionality evaluated in the product needs further study. 8.Section 2.9 (line 136), WPI instead of SPI? Response: SPI has been replaced by WPI. 9.Line 138, FESO4, indicate 4 as a subscript. Response: The number 4 has been subscripted 10.Data analysis. In order to have stronger conclusions, you must execute ANOVA and a hypothesis test for means differences to obtain information about the effect of the factors you evaluated. Something you need to include in your discussion. Response: Statistical analysis was performed using SPSS (20.0) software. Signifificant differences (p < 0.05) between means were identified using Duncan’s multiple range test. 11.Results and discussion. Line 196. Critically solution? It is not clear this information. Response: This has been revised in the revised manuscript. 12.Lines 346-348. “It was worth noting that the antioxidant…” this sentence is not clear. Please review and correct. Response: This has been revised in the revised manuscript. 13.Lines 356 and 357. “The results showed that…” This sentence is not clear. Review and correct. Response: This has been revised in the revised manuscript. 14.There is no discussion based on the statistical difference you obtained with your results (as a general comment). Response: Statistical analysis has been performed in the “Results and discussion” section of the revised manuscript. 15.Conclusions. This section can be improved; for example, in lines 364 and 365, this information is important because?.... after unfolding, the protein was more prone to be hydrolyzed? Response: Conclusions has been revised in the revised manuscript according to your suggestion. 16.Tables and Figures. Figures as in the present form are not useful to understand your results. Please change them. At least please change or modify: • Figure 4, maybe you can expose your results as a table because nor symbols nor color are helpful to observe differences among results. It is all confusion. • In figure 5, the text within the figure is not clear and very small. • Figure 6. Bars for so many treatments and results are confusing. Also, please include the result of your statistical analysis. Response: The results in Figure 4 has been exposed as Table 1 in the revised manuscript. Figure 5 and Figure 6 have been revised and statistical analysis has been performed.

Reviewer 2 Report

The authors described the effects of ultrasound treatment on the properties of walnut proteins and their hydrolysates. However, the level of novelty of this work is low as several studies have been conducted to evaluate the effects of ultrasound on the properties of walnut proteins. The work needs to be improved prior to publication. 

Comments: 

1. English language needs to be improved and polished. 
2. Lines 19 and 96-97:
     -  In the abstract, the power was mentioned to be from 600- 2000 W, while in the method part, the amplitude was mentioned. The authors shout keep it consistent. 
       - 2000 W is too high power,  Why does the authors think that such a high power is needed?

        - The Energy density (J/ml) applied during this process can be calculated to easily compare the findings with other researchers. 

3. The quality of the figures and the presentation of the obtained data should be improved. 

4. In-depth discussions are needed in the results and discussions section.

Author Response

Responses to the review comment on “High-intensity ultrasound pretreatment before enzymolysis of walnut protein isolate: effect of ultrasound on structural conformations and functional properties of the protein” (ID: molecules-1481860).

Dear editor,

Thank you very much for giving us an opportunity to revise our manuscript again, we appreciate editor and reviewers very much for their positive and constructive comments on our manuscript entitled “High-intensity ultrasound pretreatment before enzymolysis of walnut protein isolate: effect of ultrasound on structural conformations and functional properties of the protein” (ID: molecules-1481860). We have studied reviewer’s comments carefully and made revision which marked in red in the paper according to the comments.

Now we submit our revised manuscript for your kind consideration. I am the corresponding author for this paper. If you have any questions, please let me know.

I am looking forward to hearing from you soon.

Thank you very much for your kind help.

Yours sincerely,

Fei Zhao & Wentao Wang Tel: +86 53 8242850

E-mail address: feizhaozhaofei@126.com (F.Z.), wwtlxm@126.com (W.T.W.)

Reviewer: 2

  1. English language needs to be improved and polished.

Response: English language of the manuscript has been improved and polished.

  1. Lines 19 and 96-97:

  - In the abstract, the power was mentioned to be from 600-2000 W, while in the method part, the amplitude was mentioned. The authors shout keep it consistent.

   - 2000 W is too high power, Why does the authors think that such a high power is needed?

 - The Energy density (J/ml) applied during this process can be calculated to easily compare the findings with other researchers.

Response: The power of protein treatments has been converted to watts. The high power (2000 W) was researched in order to better study the effect of all power ultrasonic treatment on walnut protein structure. The watts applied during this process also can be calculated to easily compare the findings with other researchers in some literatures.

Jiang L.Z., Wang J., Li Y., Wang Z.J., Liang J., Wang R., Chen Y., Ma, W.J. Qi B.K., Zhang M. Effects of ultrasound on the structure and physical properties of black bean protein isolates. Food Research International 2014, 62, 595-601, http://dx.doi.org/10.1016/j.foodres.2014.04.022.

Resendiz-Vazquez J.A., Ulloa J.A., Urías-Silvas J.E., Bautista-Rosales P.U., Ramírez-Ramírez J.C.,

Rosas-Ulloa P. , González-Torres L. Effect of high-intensity ultrasound on the technofunctional properties and structure of jackfruit (Artocarpus heterophyllus) seed protein isolate. Ultrasonics Sonochemistry 2017, 37, 436-444, http://dx.doi.org/10.1016/j.ultsonch.2017.01.042.

Liu R., Liu Q., Xiong S.B., Fu Y.C., Chen L. Effects of high intensity unltrasound on structural and physicochemical properties of myosin from silver carp. Ultrasonics Sonochemistry 2017,37, 150-157, http://dx.doi.org/10.1016/j.ultsonch.2016.12.039.

  1. The quality of the figures and the presentation of the obtained data should be improved.

Response: The figures and the presentation of the obtained data have been improved. Statistical analysis was performed using SPSS (20.0) software.

  1. In-depth discussions are needed in the results and discussions section.

Response: In-depth discussions have been improved in the “Results and discussions” section of the revised manuscript.

Round 2

Reviewer 1 Report

The manuscript has been improved a lot. My only concern is the presentation of the data in figures 1, 4 and 5. These figures are unclear. Please re-consider to put this information in tables or make a figure with less information, and the rest of the data can be included as supplementary material. This way, you can explain better the main findings in your research. 

Author Response

Responses to the review comment on “High-intensity ultrasound pretreatment before enzymolysis of walnut protein isolate: effect of ultrasound on structural conformations and functional properties of the protein” (ID: molecules-1481860).

Dear editor,

Thank you very much for giving us an opportunity to revise our manuscript again, we appreciate editor and reviewers very much for their positive and constructive comments on our manuscript entitled “High-intensity ultrasound pretreatment before enzymolysis of walnut protein isolate: effect of ultrasound on structural conformations and functional properties of the protein” (ID: molecules-1481860). We have studied reviewer’s comments carefully and made revision which marked in red in the paper according to the comments.

Now we submit our revised manuscript for your kind consideration. I am the corresponding author for this paper. If you have any questions, please let me know.

I am looking forward to hearing from you soon.

Thank you very much for your kind help.

Yours sincerely,

Fei Zhao & Wentao WangTel: +86 53 8242850

  • mail address: feizhaozhaofei@126.com (F.Z.), wwtlxm@126.com(W.T.W.)

Reviewer: 1

The manuscript has been improved a lot. My only concern is the presentation of the data in figures 1, 4 and 5. These figures are unclear. Please re-consider to put this information in tables or make a figure with less information, and the rest of the data can be included as supplementary material. This way, you can explain better the main findings in your research.  

Response: The results in Figure 1 and Figure 5 have been exposed as Table 1, Table 3, Table 4 and Table 5 in the revised manuscript according to your suggestions. However, the original format of Figure 4 is retained to give consideration to both the presentation of data and the aesthetics of the overall layout of the paper. The shift of the maximum wavelength in the intrinsic fluorescence spectra of all the samples showed that the WPI structure had been changed after ultrasonic treatment. The change of fluorescence spectrum are also exposed as figures in some literatures of other researchers.

Wang, Y.T.; Wang, Z.J.; Handa, C.L.; Xu, J. Effects of ultrasound pre-treatment on the structure of β-conglycinin and glycinin and the antioxidant activity of their hydrolysates. Food Chem. 2017, 218, 165-172, https://doi.org/10.1016/j.foodchem.2016.09.069.

Huang, L.; Ding, X.; Dai, C.; Ma, H. Changes in the structure and dissociation of soybean protein isolate induced by ultrasound-assisted acid pretreatment. Food Chem. 2017, 232, 727-732, https://doi.org/10.1016/j.foodchem.2017.04.077.

Xiong, W.; Wang, Y.; Zhang, C.; Wan, J.; Shah, B.R.; Pei, Y.; Li, B. High intensity ultrasound modified ovalbumin: Structure, interface and gelation properties. Ultrason. Sonochem. 2016, 31, 302-309, https://doi.org/10.1016/j.ultsonch.2016.01.014.

Reviewer 2 Report

The authors have revised the manuscript carefully and the revised version can be published in this journal. 

Author Response

Responses to the review comment on “High-intensity ultrasound pretreatment before enzymolysis of walnut protein isolate: effect of ultrasound on structural conformations and functional properties of the protein” (ID: molecules-1481860).

We are very grateful for your recognition of our work, and once again thank you very much for your previous positive and constructive comments on our manuscript. I am the corresponding author for this paper. If you have any questions, please let me know.

Thank you very much for your kind help.

Yours sincerely,

Fei Zhao & Wentao Wang

Tel: +86 53 8242850

E-mail address: feizhaozhaofei@126.com (F.Z.), wwtlxm@126.com (W.T.W.)